

# Ground-based Temperature and Humidity Profiling: Combining Active and Passive Remote Sensors

David D. Turner[1] and Ulrich Löhnert[2]

[1] NOAA / OAR / Global Systems Laboratory
[2] University of Cologne / Institute of Geophysics and Meteorology

Submitted 31 August 2020

*Special Issue for the 11[th] International Symposium on Tropospheric Profiling in Atmospheric Measurement Technology*

Corresponding Author:
Dr. David Turner
NOAA Global Systems Laboratory
325 Broadway, Boulder, CO 80305
Voice: +1-303-497-6097
Email: dave.turner@noaa.gov



## Abstract

Thermodynamic profiles in the planetary boundary layer (PBL) are important observations for a
range of atmospheric research and operational needs.  These profiles can be retrieved from
passively sensed spectral infrared (IR) or microwave (MW) radiance observations, or can be
more directly measured by active remote sensors such as water vapor differential absorption
lidars (DIALs).  This paper explores the synergy of combining ground-based IR, MW, and DIAL
observations using an optimal estimation retrieval framework, quantifying the reduction in the
uncertainty in the retrieved profiles and the increase in information content as additional
observations are added to IR-only and MW-only retrievals.

This study uses ground-based observations collected during the Perdigao field campaign in
central Portugal in 2017 and during the DIAL demonstration campaign at the Atmospheric
Radiation Measurement Southern Great Plains site in 2017.  The results show that the
information content in both temperature and water vapor is higher for IR instrument relative to
the MW instrument (thereby resulting in smaller uncertainties), and that the combined IR+MW
retrieval is very similar to the IR-only retrieval below 1.5 km.  However, including the partial
profile of water vapor observed by the DIAL increases the information content in the combined
IR+DIAL and MW+DIAL water vapor retrievals substantially, with the exact impact vertically
depending on the characteristics of the DIAL instrument itself.  Furthermore, there is slight
increase in the information content in the retrieved temperature profile using the IR+DIAL
relative to the IR-only; this was not observed in the MW+DIAL retrieval.

## 1.  Introduction


High temporal resolution thermodynamic profiles in the planetary boundary layer (PBL)
are needed for a wide range of research and operational weather forecasting needs
(Wulfmeyer et al. 2015).  For example, the vertical distribution of water vapor and temperature
changes markedly over the diurnal cycle, the passage of synoptic features such as frontal
boundaries and dry lines can cause very rapid changes in the thermodynamic structure of the
PBL, and the evolution of convective weather with evaporation-driven cold pools impacts both
the temperature and humidity profiles and feeds back on the storm's evolution.  Indeed, a large
number of groups have called for improvements in the thermodynamic profiling in the PBL, and
the establishment of ground-based networks to provide these datasets to the atmospheric
science community (e.g., Dabberdt et al. 2005; NRC 2009).
Progress is being made, albeit perhaps slowly.  There are a large number of case studies
using PBL thermodynamic profiling systems to gain insight into how the convective properties
of atmosphere changes (e.g., Feltz et al. 2003; Cimini et al. 2015; Bluestein et al. 2017; Toms et
al. 2017; Mueller et al. 2017), analyses of long-time series to show the capability of these
systems (Löhnert and Maier 2012; Wagner et al. 2008), and utility for improving short-term
nowcasts and forecasts (e.g., Cimini 2011; Caumont et al. 2016; Hu et al. 2019; Coniglio 2019).
In Europe, there are a large number of microwave radiometers that are being
characterized and assimilated (experimentally) into numerical weather prediction models
(Cimini et al. 2018; De Angelis et al. 2017).  Activities in the US have focused primarily on field
campaigns, and the Plains Elevated Convection at Night (PECAN; Geerts et al. 2017) in
particular, which deployed a small network of 6 infrared spectrometers in the central US.  The
PECAN observations are being used to study a range of atmospheric phenomena both
observationally (e.g., Gasmick et al. 2018; Loveless et al. 2019) and via use in numerical weather
prediction models (Johnson et al. 2018; Degelia et al. 2019).
However, these different ground-based remote sensors have generally not been
collocated which makes evaluating the relative differences in the information content of the
observations difficult.  This paper takes advantage of two field campaigns where multiple
ground-based remote sensing systems were collocated to evaluate the relative strengths and
weaknesses of these different observations for thermodynamic profiling in the PBL.  The two
campaigns are Perdigao, which occurred in central Portugal in May-June of 2017 (Fernando et
al. 2019), and a campaign at the ARM Southern Great Plains site (Sisterson et al. 2016) in May-
June 2017 to compare a newly developed broadband differential absorption lidar for water
vapor profiling with other instruments (Newsom et al. 2020).

## 2.  Instruments


While there are many different instruments that could be included in this analysis, we
will focus on four instruments that have been demonstrated to run operationally in unattended
modes for weeks or longer, and either already are or will likely soon become commercially
available. Two of these instruments are passive remote sensors (i.e., they do not transmit
electromagnetic energy to the atmosphere) while two are active remote sensors.



### 2.1. Microwave radiometer

One type of passive thermodynamic profiler is a microwave radiometer (MWR). MWRs used for thermodynamic profiling typically have multiple channels along the high frequency side of the 22.2 GHz water vapor absorption line (i.e., from 22.2 to 31 GHz) and on the low frequency side of the 60 GHz oxygen absorption complex (i.e., from 51 to 60 GHz). Height dependent pressure broadening of the water vapor line allows the retrieval of a coarsely resolved water vapor profile, whereas temperature profile information is obtained from the frequency dependent optical depth. Generally speaking, the more transparent frequencies provide information through a deeper portion of the atmosphere and the optically thick channels provide information closer to the MWR. Oxygen is well mixed in the atmosphere and its concentration is known, thus the downwelling radiance observed in the channels that are primarily sensitive to oxygen can be used to infer the temperature profile. Water vapor concentration profiles can be determined from the channels that have sensitivity to water vapor after the temperature profile is known. However, there is some level of absorption due to oxygen in the 22-31 GHz range and water vapor in the 51-60 GHz range, so retrieval methods need to account for this 'cross-talk', and provide some estimate of the correlated errors in the retrieved profiles.

For this study, we used a 14-channel Humidity and Temperature Profiling (HATPRO) microwave radiometer (Rose et al. 2005). This is a fourth-generation system, which is part of the Collaborative Lower Atmospheric Mobile Profiling System (CLAMPS; Wagner et al. 2019). The instrument specifications are given in Table 1. The radiometric uncertainty in these observations were determined via a time-series analysis of the observed brightness temperatures when the atmosphere could be assumed to be quasi-stationary. These values are provided in Table 1. These radiometric uncertainties are assumed to be uncorrelated between the different channels.

### 2.2. AERI

The second passive remote sensor studied here is the Atmospheric Emitted Radiance Interferometer (AERI). The AERI is a Fourier transform spectrometer designed to measure infrared radiation emitted by the atmosphere between 3.3 and 19 μm in wavelength (3000 to 520 cm$^{-1}$) with a spectral resolution of 0.5 cm$^{-1}$. The AERI was designed specifically for the Department of Energy's Atmospheric Radiation Measurement (ARM) program (Turner et al. 2016a; Knuteson et al. 2004 a,b). Its specifications can also be found in Table 1.

The radiometric uncertainty in the AERI observations is derived from the imaginary component of the AERI's calibration equation (Revercomb et al. 1988), and thus the noise spectrum can be derived for each sky observation period. Turner and Blumberg (2019) have demonstrated that the radiometric noise in the AERI observations is spectrally uncorrelated.

### 2.3. NCAR water vapor DIAL

Water vapor differential absorption lidar (DIAL) work by transmitting pulsed laser energy at two wavelengths, one of which is selected to have markedly higher water vapor absorption than the other. These two frequencies are typically referred to as the on-line and off-line

frequencies. If the two wavelengths are spectrally close to each other (e.g., within a nm in
wavelength), then many of the terms that describe the ratio of the strength of the
backscattered signals cancel out. The ratio of the on- to off-line return signals is directly related
to the water vapor concentration profile.
The National Center for Atmospheric Research (NCAR) has developed a micropulse water
vapor DIAL. The approach used by this lidar is the so-called "narrowband DIAL" approach
wherein the laser emits monochromatic pulses of energy. Thus, because the characteristics of
the absorption line are well known, the method is self-calibrating and no external calibration
source is needed. Narrowband DIAL systems require extremely high spectral purity in the
outgoing laser, as subtle changes in the wavelength (especially for the on-line channel) even for
a small number of laser pulses in the averaging window can introduce biases in the derived
water vapor profile because the incorrect absorption cross-section is used in the derivation.
The laser in the NCAR DIAL, henceforth called the nDIAL, emits low pulse energies at high
pulse repetition rate (Spuler et al 2015). The outgoing laser beam is expanded by a portion of
the primary telescope, which makes the lidar system eye-safe. The nDIAL system has its origins
at Montana State University (MSU), wherein commercially available laser diodes developed for
telecommunications were used as the laser source (Nehrir et al. 2012), and MSU continues to
collaborate with NCAR to advance the nDIAL technology. A single photon counting detector is
used to detect the backscattered signals in both the on-line and off-line channels. High
transmission, narrowband interference filters are used to reject energy (e.g., solar background)
outside the desired frequency range of the desired signals. The technical details of this system
are provided in Table 1.
The signal-to-noise ratio (SNR) in DIAL systems is strongly dependent upon the strength of
the backscattering signal as a function of range. Aerosol particles provide an efficient scattering
source, and because aerosol concentration decreases markedly above the top of the PBL, the
SNR also drops sharply above this level. However, the actual range wherein the lidar makes
good water vapor measurements is a function of the pulse energy, the efficiency of the
detector system (e.g., size of the telescope, transmission of the detection optics, sensitivity of
the detector), and the vertical profiles of both the aerosol and water vapor concentrations. For
this study, the backscattered photon data were coadded for 1-minute before deriving the water
vapor profile.
Virtually all lidar systems have difficulties accurately measuring atmospheric properties
close to the lidar itself. Ultimately, this is due to a mismatch between the outgoing laser beam
and the detector and leads to a systematic error that varies with height. This systematic error
reduces to zero at some range, and the region were the error is nonzero is referred to as the
"overlap" region. For many lidar systems, an empirically determined correction can be applied
to reduce the maximum range of the non-zero overlap error. For the current version of the
nDIAL, approximately the lowest 500 m suffers from a varying overlap correction (S. Spuler,
personal communication), and thus is not used in this analysis.
The uncertainty in the nDIAL observations is directly calculated by assuming that the
detected backscatter signal follows a Poisson distribution, and propagating the uncertainty in
the backscatter profile through the DIAL equation. A similar approach was used for the SGP
Raman lidar, and the noise estimate derived from Poisson statistics agrees with that derived
using an autocovariance analysis (Turner et al. 2014).





The nDIAL has been deployed in a number of different field campaigns.  In particular, the
water vapor profile observed by the nDIAL have been compared to water vapor profiles
measured by radiosondes and independently retrieved from collocated AERI and MWR systems
(Weckwerth et al. 2016). These comparisons demonstrate that the nDIAL agrees well with these
other sensors (e.g., the bias error relative to radiosondes is less than 0.3 g/m$^3$) and has no
significant day vs. night differences in sensitivity (e.g, due to solar background).  In 2018, NCAR
constructed 4 additional units (bringing the total number of nDIAL systems to five), which were
deployed in a network configuration at the Department of Energy's Atmospheric Radiation
Measurement (ARM) Southern Great Plains site (SGP, Sisterson et al. 2016) from April through
July 2019.
*2.4. Vaisala water vapor DIAL*
Vaisala is also developing a micropulse water vapor DIAL (henceforth called the vDIAL).  This
lidar system is based upon the CL51 ceilometer design; this ceilometer is used operationally
around the world.  Unlike the nDIAL, the vDIAL transmits a spectrally broad pulse of laser
energy that encompasses several water vapor absorption lines ("on-line channel") and in a
nearby spectral window with no absorption lines ("off-line").  This approach is less technically
demanding on the laser specifications (e.g., the requirement for high spectral purity is much
smaller), but the tradeoff is that the measurement is no longer self-calibrating (Newsom et al.
2020).  For this particular broadband DIAL implementation, the reference measurement is a
well-calibrated surface level in-situ sensor integrated into the DIAL, and measurements from
this sensor are used in an iterative retrieval approach to derive the water vapor profile
(Newsom et al. 2020).
The vDIAL actually consists of two independent broadband DIAL systems integrated
together. The first system has a wide field-of-view, thereby resulting in a very small overlap
region and allowing the lidar to profile water vapor down to 50 m above ground level (AGL).
However, this wide field-of-view results in additional solar background photons and the SNR
decreases very rapidly with range.  The second system has a much narrower field of view, which
results in a deeper overlap region but also enables the lidar to profile water vapor much higher.
Cross-talk between the two independent systems is eliminated by operating one system for 5-s,
and then operating the other for the next 5-s.  The water vapor profiles are derived
independently for the wide and narrow field-of-view systems, and then they are merged
linearly between 300 and 400 m.  Additional details on this system are provided in Newsom et
al. (2020).
The vDIAL system uses analog detection, and thus the uncertainties in the backscatter do
not follow a Poisson distribution like in the nDIAL.  Instead, the uncertainties in the vDIAL water
vapor profile are estimated by deriving water vapor profiles every 2-minutes, and computing
the standard deviation from these data at each height across a 20-minute window to provide
the uncertainty in the standard 20-min average water vapor profile.
The vDIAL system was deployed to the ARM SGP in May-June 2017, where it was compared
against water vapor profiles observed by the ARM Raman lidar (Turner et al. 2016b; Turner and
Goldsmith 1999), radiosondes, and retrieved from the AERI.



## 3. Retrieval algorithm

Passive spectral radiometers, such as the MWRs and AERIs, measure radiance, and thermodynamic profiles must be retrieved from these observations. However, this is an ill-posed problem, as there could exist multiple solutions (e.g., different thermodynamic profiles) that would yield the observed radiance. Thus, the retrieval algorithm must incorporate additional information to constrain the solution to a potentially valid solution. Here, we have elected to use the optimal estimation approach (Rodgers 2000; Maahn et al. 2020), which is a 1-dimensional variational method. We have modified the AERIoe optimal estimation retrieval algorithm (Turner and Löhnert 2014) to use AERI and/or MWR data, together with *a priori* dataset that specifies how temperature and humidity covary with height, as input. This algorithm has already been modified to include additional observations, such as water vapor lidars (Turner and Blumberg 2019), and thus in these cases the retrieval is finding the temperature and humidity profiles that satisfies both the observed radiance and the (partial) profile of water vapor observed by the DIAL simultaneously.

We desire to retrieve the thermodynamic profile $X$ (i.e., both the temperature and humidity profile, so $X = \left[ [T_1, T_2, \ldots, T_p]^T, [q_1, q_2, \ldots, q_p]^T \right]$ where $T_i$ and $q_i$ are the temperature and water vapor mixing ratio in the i[th] vertical bin. We will refer to $X_n$ as the state vector on the *n*th iteration. The observations from the AERI, MWR, and DIALs will form the observation vector $Y$. A forward model $F$ is used to compute a pseudo observation $F(X)$, which is then compared with $Y$. If they disagree, then the state vector is modified to provide a new estimate $(X_{n+1})$ following

$$X_{n+1} = X_a + (\gamma S_a^{-1} + K_n^T S_\epsilon^{-1} K_n)^{-1} K_n^T S_\epsilon^{-1} \left( Y - F(X_n) + K_n(X_n - X_a) \right) \qquad \text{(Eq 1)}$$

where $K$ is the Jacobian of $F$, $X_a$ is the mean *a priori*, and $S_a$ is the covariance matrix of the *a priori* dataset (see Section 3.2). $S_\epsilon$ denotes the combined forward model and observation error covariance matrix. The observation error for the single instruments is considered as described in the subsection of Section 2 and the forward model uncertainty is discussed in Section 3.1. The superscripts [T] and [-1] denote matrix transpose and matrix inverse, respectively. Because $F$ is moderately non-linear in $X$, optimal estimation is formulated as an iterative method, where the subscript $n$ indicates the iteration number; for our studies, we typically start with $X_0 = X_a$. The scalar $\gamma$ is used to stabilize the retrieval when *n* is small to improve the convergence rate and decreases to unity as *n* increases; the description on how $\gamma$ is used is explained in Turner and Löhnert (2014). Note that due to the non-linearity of the forward models applied for the microwave and infrared radiative transfer, the Jacobians are required to be recomputed for each iteration. We continue to iterate Eq 1 until

$$\left( F(X_{n+1}) - F(X_n) \right)^T (K_n S_a K_n^T + S_\epsilon)^{-1} \left( F(X_{n+1}) - F(X_n) \right) \ll m \qquad \text{(Eq 2)}$$

where *m* is the dimension of $Y$.





### 3.1. Forward models

As shown by Eq 1, a forward model is needed to transform the current state vector $X_n$ into the observational domain so it can then be compared to the observation vector $Y$. In this study, four different forward models are used (one for each instrument).

For the passive radiometers, the forward models are line-by-line radiative transfer models. The monochromatic MonoRTM radiative transfer model (Clough et al. 2005; Payne et al. 2011) is used to simulate MWR observations, and the line-by-line radiative transfer model LBLRTM (Clough et al. 1995; Mlawer and Turner 2016) is used to simulate the AERI. In the latter, the monochromatic spectra are convolved with a tophat function in the time domain and then transformed to the spectral domain via a Fourier transform; this applies the AERI's lineshape function to the calculation. The vertical grid used in these calculations is specified by the *a priori* data. The pressure profile is computed from the temperature and humidity data from the current state vector using the hypsometric equation. The spectral regions used in the retrieval are given in Table 1. In the infrared, many trace gases have absorption bands, and while the spectral regions used in the retrieval are primarily sensitive to water vapor and carbon dioxide (where the latter provides the sensitivity to temperature), there are minor contributions to the downwelling radiance by other gases. We utilize the US Standard Atmosphere to provide profiles of these other trace gases for this study, but our results are insensitive to this choice.

To incorporate the DIAL data into the Eq 1, a forward model is needed for each lidar also. The purest forward model would simulate the profiles of backscatter energy that would be observed in both the on- and off-line channels for a given water vapor profile. We have elected to use the derived water vapor concentration from each lidar in the observation vector. This results in a trivial forward model for each lidar: essentially, the forward model just converts water vapor mixing ratio to water vapor number concentration for the nDIAL. The output of the vDIAL is water vapor mixing ratio, so that forward model is just the unity function.

### 3.2. The a priori dataset

There has been only one campaign that had an AERI, HATPRO, and water vapor DIAL collocated with each other: the Perdigao campaign that was held in Portugal from 1 May to 15 June 2017 (Fernando et al. 2019). We specified a 48-level vertical grid for the retrievals, starting at 0 m above ground level (AGL), the next level at 10 m AGL, and each subsequent height bin is 10% thicker than the previous one. Although ~150 radiosondes were launched during Perdigao, these are not enough to accurately compute the level-to-level covariance for the 96-element state vector (i.e., $X$ has 48 levels for temperature, and 48 for water vapor). Therefore, we used 1571 radiosondes launched in the months of April, May, June, and July over the last decade by the Portuguese weather service at Lisbon to compute $X_a$ and $S_a$. This *a priori* information was used in all of the retrievals shown here.

The vDIAL was not part of the Perdigao deployment, so we are using AERI and vDIAL data collected between 15 May to 12 June 2017 at the SGP site instead. Both the Perdigao and SGP datasets used here were collected in the spring, but the SGP climatology is different than that in Portugal necessitating the use of a different *a priori* dataset. We have used over 2000 radiosondes launched at the SGP during the months of April, May, and June over the past decade to derive the *a priori* information for this site.

### 3.3. Characterizing the information content in the retrieved profile

One advantage of the optimal estimation framework is that the uncertainties in the retrieval, which includes contributions from both the uncertainties in the observations and *a priori* as well as the sensitivity of the forward model, is a direct output of the framework. If the "optimal" solution is $X_{op}$, which is the solution after both $\gamma = 1$ and Eq 2 indicates that the solution has converged after $nc$ iterations, then the covariance of the optimal solution is given by

$$S_{op} = (S_a^{-1} + K_{nc}^T S_\epsilon^{-1} K_{nc})^{-1} \qquad \text{(Eq 3)}$$

We will look at the square root of the diagonal elements of $S_{op}$ to quantify how the 1-σ uncertainties of the retrieved profiles change as different instrument combinations are used in the observation vector.

A second advantage of this method is that the averaging kernel $A$ provides a direct estimate of the sensitivity of the retrieved profile at each height to perturbations at that height. This matrix is computed as

$$A = (S_a^{-1} + K_{nc}^T S_\epsilon^{-1} K_{nc})^{-1} K_{nc}^T S_\epsilon^{-1} K_{nc} = I - S_{op} S_a^{-1} \qquad \text{(Eq 4)}$$

The diagonal components of $A$ provides the degrees of freedom for signal (DFS; Rodgers 2000) for each height in the retrieved profile. If the observations had very high information content at each level of the retrieved profile, then the diagonal elements of $S_{op}$ would be small relative to the diagonal elements of the *a priori*, and thus the trace of $A$ would approach the dimension of $X$. The total DFS, which is equal to the trace of $A$, provides a metric for how many independent pieces of information exist in the observation.

For this study, we recognize that the matrices $A$, $S_{op}$, and $S_a$ really have four equal sized quadrants that correspond to

$$\begin{bmatrix} (T,T) & (T,q) \\ (q,T) & (q,q) \end{bmatrix}$$

We will look at the portions of $A$ and $S_{op}$ that correspond to (*T,T*) and (*q,q*) independently. Furthermore, as we will see, the DFS is typically much smaller than unity, so we will look at the profile of the cumulative DFS (cDFS), as this allows us to quickly determine how many independent levels are below some specified height, which is advantageous when talking about where in the vertical the different instruments provide sensitivity to changes in temperature and water vapor.

We want to highlight that even though lidars make explicitly range resolved measurements, their information content in the derived water vapor profile is not the same as the number of range bins. The actual information content at height z depends strongly on the noise level of the observation there. Even direct derivations of water vapor from lidar signals would benefit from being cast into a retrieval framework like what we've specified in Eq 1 because then the *a priori* information could be used to constrain the derived water vapor when the instrument's SNR decreases (e.g., Sica and Haefele 2016).

## 4. Results

Several studies have demonstrated that ground-based thermodynamic retrievals in the PBL using only AERI observations have 2-4 times larger total DFS in both temperature and water


vapor than retrievals that use only microwave data (Löhnert et al. 2009; Blumberg et al. 2015;
Wulfmeyer et al 2015). However, what is not known is how the information content changes
when partial profiles of water vapor from a differential absorption lidar (since the DIAL
observations extend only from the top of the region where full overlap is achieved to a height
where its SNR becomes small) are included in a retrieval considering the synergy of AERI, MWR,
and nDIAL or vDIAL. For example, does including a partial water vapor profile in the retrieval
result in AERI+DIAL and MWR+DIAL having equivalent cDFS for water vapor? Does including a
partial water vapor profile in a simultaneous retrieval of T(z) and q(z) (as we are doing here in
Eq 1) improve the temperature profile in any way?
In order to answer these questions, we performed eight sets of retrievals using data from
the Perdigao field campaign in Portugal (Table 2): four were using passive-only measurements
(MWRz, MWRzo, AERI, and AERI+MWRz), and four included the nDIAL together with those
passive measurements. "MWRz" denotes the case when only zenith-pointing MWR brightness
temperature observations were used in the retrieval, whereas "MWRzo" denotes the case were
both zenith and off-zenith (i.e., "oblique" elevation scans) are used. Crewell and Löhnert (2007)
demonstrated that adding elevation scan observations at frequencies where the atmosphere is
optically thick, and assuming horizontal homogeneity of the PBL, resulted in a marked increase
in the information content and hence accuracy of the retrieved temperature profile. However,
only observations made at frequencies above 55 GHz are used in these elevation scans. Even at
low elevation angles, frequencies channels below 55 GHz are too transparent and thus the
assumption of horizonal homogeneity fails very frequently (Crewell and Löhnert 2007).
As the vDIAL will soon be the first commercially available DIAL instrument for water vapor
profiling (H. Winston, personal communication), a major objective is to evaluate how including
this lidar dataset with passive observations changes the information content in the retrieved
profiles. In addition, we show the impact of the vDIAL relative to the nDIAL on our retrievals.
However, vDIAL (ARM SGP) and nDIAL (Perdigao) observations are only available at different
locations with different *a priori* datasets. In order to overcome this issue, the comparisons were
carried out in relation to the AERI instruments, which operated at both sites. The comparison of
the AERI-only from ARM-SGP and Perdigao allows us to characterize the impact of the prior on
the retrievals, since the two AERI instruments deployed in Portugal and at the SGP site have
similar error characteristics (not shown). Ultimately, we have looked at the differences
between the AERI-only and AERI+xDIAL retrievals (where x is either "v" or "n") at the two sites.

### 4.1. Case study example

To illustrate the differences between the various passive-only and passive+active retrievals,
we selected a case during Perdigao on 15 May 2017 at 05:07 UTC. This is a clear sky event, and
is representative of the retrieval quality during the entire field campaign. Figure 1 shows the
retrieved temperature (panel A) and water vapor mixing ratio (WVMR, panel B), and the
associated 1-$\sigma$ uncertainties of each (panels C and D, respectively) derived from the square root
of the diagonal of the retrieval error covariance $S_{op}$. The black line in panels A and B denote
the coincident radiosonde, whereas the other colors denote the different passive-only
retrievals.
All three passive-only retrievals (MWRzo, AERI, and AERI+MWRzo) identify the surface-
based inversion, although the retrievals that include the AERI capture it more accurately (Fig



1A).  Furthermore, the retrievals that include the AERI are able to better match the radiosonde
temperature observations above 1.5 km, whereas the MWRzo retrieval is showing a bias at
those altitudes. None of the three retrievals are able to capture the small-scale variability in the
vertical observed by the radiosonde due to the relatively coarse vertical resolution of the
retrievals.  The uncertainties in the MWRzo temperature retrievals are about 50% larger (or
more) over the lowest 3 km relative to the AERI retrievals (Fig 1C), which agrees qualitatively
with the differences to the radiosonde seen in Fig 1A.
The water vapor retrievals (Fig 1B) show two basic vertical patterns: the MWRzo retrieval is
markedly drier than the radiosonde below 1 km, whereas the AERI and AERI+MWRzo retrieval
starts dry, then becomes too wet (between 500 and 1000 m), and then becomes drier than the
radiosonde above 1500 m.  Interestingly, the nDIAL water vapor profile is also drier than the
radiosonde below 1500 m, and agrees better with the MWRzo profile.  However, the retrievals
that use the AERI data have markedly smaller uncertainties than the MWRzo below 1.5 km;
above that height, the uncertainty in the MWRzo is smaller than the AERI, although the
AERI+MWRzo retrieval has the smallest uncertainties over the entire lowest 3 km as would be
expected for a variational retrieval method.
Including the nDIAL data above 500 m into the retrieval, and thus finding a solution that
simultaneously fits both the observed radiance and the partial WVMR profile of the DIAL within
their uncertainties, yields the results shown in Fig 2. The largest impact, not surprisingly, is on
the retrieved water vapor profile (Fig 2B).  The inclusion of the nDIAL data forces the retrievals
that also include the AERI to reduce the amount of water vapor between 500 and 1000 m
(where the AERI-based retrievals were too wet in Fig 1B), which has the impact of increasing
the amount of water vapor in the AERI retrievals below 500 m (Fig 2B), resulting in the
AERI+nDIAL and AERI+MWRzo+nDIAL agreeing much better with the radiosonde.  Between 800
and 1500 m, the MWR+nDIAL retrieved profile is essentially the same as the nDIAL profile,
suggesting that the MWR is not adding any significant information to the DIAL's observation.
The impact of the nDIAL data on the water vapor uncertainty profiles can clearly be seen in Fig
2D, where all retrievals have the similar uncertainty above about 800 m where the DIAL data
are being used.  Including the DIAL data into the retrievals has a minor impact on the retrieved
temperature profiles, as all three seem to agree a bit better qualitatively with the radiosonde
above 1000 m (comparing Fig 2A with Fig 1A), and the 1-$\sigma$ uncertainties in temperature are
slightly smaller (Fig 2C with Fig 1C).

### 411    4.2. Comparing mean uncertainty profiles

While the case study above may be representative, the quality of a retrieval (i.e., its
uncertainty and information content) is case specific.  To provide a more complete picture of
how the different passive-only and active+passive retrievals compare, we computed the mean
1-$\sigma$ uncertainty profiles from all of the retrievals performed during Perdigao, as a wide range of
environmental conditions (e.g., the surface temperature ranged from approximately 9 to 33 °C
and the precipitable water vapor from 1.1 to 3.1 cm) were observed during the 5-week
campaign.  Figure 3 shows these mean uncertainty profiles for temperature (left) and water
vapor (right) for the different passive-only (solid lines) and active+passive (broken lines), and
Table 2 provides the mean values at 3 different heights.

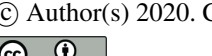



Considering the passive-only retrievals, combining the AERI and MWR together has little
impact on the resulting temperature retrieval in the lowest 3 km or on the water vapor retrieval
below 1.5 km, compared to the AERI-only retrieval.  However, the MWRz and MWRzo
outperform the AERI for water vapor above 2 km. Most strikingly, the benefit of the passive
retrieval synergy can be seen for water vapor above 1.5 km, where the improvement is up to
30% compared to the single sensor retrievals. Adding the elevation scanning data to the MWR
retrieval (i.e., the MWRzo vs MWRz) results in a smaller uncertainty in the temperature profile,
especially below 400 m.
Including the nDIAL data into the retrievals greatly reduces the 1-$\sigma$ uncertainty in the water
vapor profiles for all active+passive retrievals (relative to the passive-only results), and results
in a slight decrease in the temperature uncertainty also.  The AERI-based retrievals show
smaller uncertainties than the MWR-based retrievals, with the exception in the water vapor
retrievals above 2 km where the MWR-based retrieval has a smaller uncertainty than the AERI
retrieval.  The uncertainty in the AERI+nDIAL water vapor retrieval between 500 m and 2 km,
where the nDIAL data are used, is slightly smaller than the uncertainty in the MWRz+nDIAL
retrieval, suggesting that the AERI is adding more information to the DIAL observations than the
MWR. However, above 2 km the combination of all sensors has distinguishably the best
performance, indicating that all instruments are contributing to the sensor synergy. In
quantitative numbers, the WVMR can be retrieved via sensor synergy with accuracies between
0.4 and 0.6 g kg$^{-1}$ in the lowest 3 km, which between 1 and 2 km (the region where DIAL shows
its optimal performance), is an uncertainty reduction of up to 50% compared to the passive
retrieval synergy.

### 4.3. Comparing mean cDFS profiles

The optimal estimation framework used in this study uses the *a priori* to help constrain the
ill-posed retrieval, thereby allowing the algorithm to converge to a realistic solution more
frequently.  Looking at the DFS profile, especially when summed with altitude from the surface
(called here the cumulative DFS profile), enables one to understand where the independent
data in the observations are located vertically.  Figure 4 shows the mean cumulative DFS
profiles for the different retrievals; mean values at three specific heights are provided in Table
3.
There are several important features in this figure.  First, adding the elevation scanning data
to the MWR retrieval (i.e., comparing the MWRz-only vs. MWRzo-only) increases the total DFS
for temperature at 3 km by 0.4 (from 2.15 to 2.57), with almost all of this increase in the first
500 m.  [Note, however, that we have only used a single elevation angle to the MWRzo (Table
1), and the inclusions of additional elevation angles would result in a slight increase the cDFS for
temperature.] The AERI-only temperature retrieval has more information (3.87) in the lowest
500 m than the MWRzo-only retrieval has in the lowest 3 km (2.57).  Most of the information in
the temperature retrievals is below 1.5 km, as the cDFS profiles become relatively constant
above that level; this suggests that these passive-only and active+passive temperature
retrievals will have limited ability to retrieve the structure of the temperature profile above
that height.
The passive-only retrievals of water vapor show less total DFS (using the value at 3 km
height) during Perdigao relative to datasets at other field campaigns (e.g., Turner and Löhnert



2014; Blumberg et al. 2015).   This is likely due to the spread in the covariance of the prior,
because if the prior had (hypothetically) negligible spread then the derived information content
from the observations would be vanishingly small.  Nonetheless, we can still use this prior to
demonstrate how the addition of the DIAL data to the retrievals changes the information
content.  The cDFS profiles for the water vapor retrievals clearly show the impact of including
the nDIAL data above 500 m, as the cDFS profiles for the active+passive retrievals are markedly
larger above that height than the passive-only retrievals (i.e., with values between 6 to 7
compared to 2 to 3 at 3 km).  The additional information on water vapor in the AERI below 500
m relative to the MWR is clearly seen.  However, the lidar does not always provide data to the
same altitude and its noise levels can depend on atmospheric conditions (e.g., if there is a cloud
above the lidar or not), and thus the spread in the cumulative DFS profiles was quite large (e.g.,
from 2.0 to 9.4 for the MWRz+nDIAL at 3 km height; Table 3).

### *4.4. Impact of clouds*

One of the often-stated advantages of MWR-based retrievals, relative to infrared-based
retrievals, is the ability to profile through clouds because the optical thickness of the cloud is
markedly smaller in the microwave relative to the infrared for a given liquid water path (LWP).
Figure 5 shows cDFS profiles from the MWRz-only and AERI-only temperature and water vapor
retrievals during a 2h period when the sky transitioned from virtually clear sky to overcast.
Three profiles with different LWP amounts (2, 10, and 60 g m$^{-2}$, where the infrared is essentially
opaque for the last – Turner 2007) are shown.  The cloud base was at 1100 m and was assumed
to be 100 m thick (there was no way to determine cloud top from other observations at the
site).  First, notice that as the cloud becomes optically thicker, the retrievals have more
information about the temperature at cloud base.  Second, the cloud becomes opaque in the
infrared quickly, hence the cumulative DFS profile becomes essentially constant (especially for
water vapor) above the cloud as the LWP values approach 60 g m$^{-2}$.  Meanwhile, the cloud is
semi-transparent in the microwave for all LWP values, which is seen by the increasing cDFS
profile (especially for water vapor) above the cloud.  However, there is still only a small amount
of information in the observations at heights above 1 km in the MWR (see left-hand panel of Fig
4), and thus the increase in the information content in the MWR retrieval above the cloud is
relatively limited.
The accurate understanding of where the information exists vertically is useful in order to
properly assimilate these profiles into a numerical weather prediction model.  There is often
significant level-to-level correlation in the uncertainties of profiles retrieved from passive
remote sensors (e.g., see Figure 10 of Turner and Blumberg 2019), and most data assimilation
systems are not yet configured to handle correlated error in the observations.  Coniglio et al.
(2019) used the cDFS profile to identify the heights that should be assimilated to minimize the
amount of correlated error from the retrieved profiles.  Starting at a specified height (e.g., 50
m), they identified heights where the cDFS had increased by 1 above that height, and this
process continued until they either were unable to identify any other points or had reached the
maximum height that they wanted to assimilate.  This is illustrated by the dots on the profiles in
Fig 5, with the first height taken at 50 m.  For the AERI-retrieved profiles, three levels would be
assimilated below the cloud with an additional level at cloud base or just above; the height of
all of the temperature levels is pretty consistent for these three profiles.  For the MWR, only
two levels would be assimilated due to the lower information content in the microwave
observations, with the height of the second point changing dramatically due to how the cloud
influences the vertical distribution of the DFS profile. Again, we remind the reader that the
total DFS seen in this example is lower than that seen using this same retrieval framework in
other field campaigns; we attribute this to the lack of spread in the *a priori* dataset used at
Perdigao.

### 513    *4.5. Sensitivity to the nDIAL vs. vDIAL*

The impact of adding any new observation depends partially on its error covariance matrix,
as observations with larger uncertainties will add less information to the retrieved profile than
observations with smaller uncertainties. For many lidars, coadding photon counting data in
either time or altitude reduces the random errors, and thus would increase the information
content and impact of using these lidar data in retrievals such as these. However, other
features of the observations are also important. For example, during Perdigao, the lowest
range gate that was considered useful from the nDIAL was at 500 m; data below that level
suffered from systematic errors associated with the overlap function of the lidar (S. Spuler,
personal communication). However, the vDIAL was designed to make good measurements at
50 m above the surface, although generally speaking its maximum range is much less (order 1
km; Newsom et al. 2020) than the nDIAL system (which frequently makes good water vapor
measurements to altitudes well above 2 km). A natural question is how would the results
already shown change if the vDIAL system was used instead of the nDIAL?
Unfortunately, this isn't straight-forward to answer as the vDIAL was not collocated with the
other Perdigao instruments. Instead, we use the 6-week deployment of the vDIAL at the ARM
SGP site (Newsom et al. 2020), which has an AERI with similar noise characteristics as the AERI
deployed at Perdigao, as a surrogate. However, different *a priori* datasets were used for the
retrievals at the two sites, which impacts the retrievals and hence the analysis. To help adjust
for the contribution of the two priors, we performed AERI-only retrievals and AERI+vDIAL
retrievals at the SGP so that we could look at the difference between the two, and compare
that to the difference between the AERI-only and AERI+nDIAL retrievals at Perdigao (Figure 6).
The impact of the vDIAL data on the water vapor retrieval is most significant between 300
and 1500 m and reaches relative values of up to 50% uncertainty reduction compared to the
AERI-only retrieval. Above 1500 m, the AERI+vDIAL WVMR uncertainties increase quickly with
height and approach the AERI-only uncertainties at 3 km. The AERI+nDIAL uncertainties are
very similar to the AERI-only below 500 m (because the nDIAL data is not available at those
levels), but are approximately 2x smaller than the AERI-only for all height between 500 m and 3
km. Further, the change in the cDFS between 500 m and 3 km is larger for the nDIAL system
relative to the vDIAL (Table 3). Thus, the ability of the nDIAL to see deeper into the troposphere
than the vDIAL is clearly shown. Interestingly, the water vapor uncertainty in the AERI+vDIAL is
smaller than the AERI+nDIAL in the 500 to 900 m range; however, this could easily be changed
by adjusting how the DIAL data were coadded in the nDIAL (which had 1-min temporal
resolution relative to the 20-min temporal resolution of the vDIAL – see Table 1).
Perhaps most noteworthy is the relative impact of the two DIALs on the retrieved
temperature profile. The addition of the vDIAL data has almost no impact on the uncertainty or
the cDFS profile relative to the AERI-only (Fig 6, Tables 2 and 3), whereas the nDIAL has a

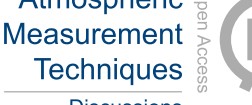

marked impact on the retrieved temperature profile in the range from 500 m to 2.5 km with an
reduction of the uncertainty of up to 0.25 K compared to the AERI-only retrieval. Here, the
instrument synergy is obtained through a more exact determination of the water vapor profile
by the nDIAL, which enables the AERI to reach a higher DFS value for temperature.
## 5. Conclusions

Many applications require profiles of temperature and humidity in the PBL. However, the
accuracy and information content from different ground-based remote sensing instruments is
not the same. Previous work (e.g., Löhnert et al. 2009; Blumberg et al. 2015) demonstrated
that there is more information content in both temperature and water vapor from spectral
infrared measurements (such as made by the AERI) than in spectral microwave radiometer
measurements. These results depend strongly on the characteristics of the instrument systems
being used; for example, if future generation MWRs are improved to have smaller random
errors, then the information content in the observations would increase. The on-line python
modules provided by Maahn et al. (2020) can be used to explore how the information content
would change for different assumed random error levels in the MWR.
This study investigated the impact of ground-based sensor synergy for PBL thermodynamic
profiling, and in particular, how the information content and random errors would change if an
active remote sensor such as a water vapor DIAL was included into the retrieval. An open
question going into this research was whether the inclusion of the water vapor DIAL
observations with MWR radiance observations would have the same information content as
retrievals that used the DIAL with the AERI observations. An important aspect of this study is
that the same *a priori* data and retrieval framework were used for all of the different retrievals
shown in this paper, which is crucial to truly quantify the differences as different retrieval
frameworks can result in markedly different retrievals (Maahn et al. 2020). We have shown
that including the DIAL data increases the water vapor information content and reduces water
vapor errors in both the AERI+DIAL and MWR+DIAL retrievals, relative to the passive-only
retrievals. However, the AERI+DIAL continues to have more information on water vapor than
the MWR+DIAL. The best retrieval performance is observed when all three instruments are
combined in one retrieval. Improvements are shown that decrease the uncertainty by 50%
compared to passive-only retrievals between 1 and 2 km. At Perdigao, the AERI is shown to
dominate retrieval accuracy in the lowest 500 m, from 500 m to 2 km it is the DIAL that
primarily determines the accuracy, and above 2 km the three instruments complement each
other optimally to obtain the best solution. Furthermore, the addition of the water vapor DIAL
observations (slightly) improves the information content in temperature retrievals from the
AERI+DIAL, but has no impact on the temperature profiles for the MWR+DIAL.
Passive ground-based remote sensors are relatively common, as these technologies are
more mature, have been commercially available for several decades, and have been operated
in networks (e.g., Caumont et al. 2016; Geerts et al. 2017; Yang and Min 2018). The recent
advances in water vapor DIAL (e.g., Spuler et al. 2015; Newsom et al. 2020) are leading to the
possibility that the two DIALs used in this study could be commercially available in the next
several years, which is why they formed the focus of this study. There are other
thermodynamic profiling active remote sensors that could be combined with MWRs and AERIs:



for example, Raman lidar and Radio Acoustic Sounding Systems (RASS). Studies have been
conducted combining Raman lidar with both MWR data (e.g., Barrera-Verdejo et al 2016; Foth
and Pospichal 2017) and AERI data (e.g., Turner and Blumberg 2019); however, these studies
were in different environments using different *a priori* datasets, which makes quantitatively
comparing their accuracy and information content problematic. There are currently efforts
underway to evaluate the impact of RASS virtual temperature profiles observations on both
AERI and MWR observations.
Sensor synergy does not have to just involve ground-based sensors. Ground-based MWR
and AERI observations can also be combined with satellite observations to improve information
content and accuracy, especially in the middle- and upper troposphere. Feltz et al. (2003)
showed the impact on AERI retrievals and how these improved profiles could be used for
evaluating thermodynamic structure near storms, while Ebell et al. (2013) performed a more
classical information content study. Additional efforts (e.g., such as Toprov and Löhnert 2020)
are needed, which show the impact of the high-temporal and high-spectral resolution
geostationary infrared sounders with ground-based remote sensing systems and the impact on
stability indices and other parameters.
It is possible that readers will consider this study as a suggestion about the optimal ground-
based solution for thermodynamic profiling, especially for future operational networks. This
paper provides insights into only one aspect of the cost-benefit solution (i.e., the relative
differences of information content); considerations as to ease of use, durability and hardiness,
calibration stability, and other scientific traits (e.g., does the instrument provide information on
macro- or microphysical cloud properties, aerosol properties, trace gases, etc.) also need to be
considered.

## Acknowledgments

This research was supported in part by the Department of Energy's Atmospheric System
Research (ASR) program (DE-SC0014375 and 89243019SSC000034) and NOAA's Atmospheric
Science for Renewable Energy program. We thank the groups that helped to collect the two
primary datasets from the Perdigao field campaign in used in this paper: Petra Klein, Elizabeth
Smith, Josh Gebauer, and Tyler Bell at the University of Oklahoma; and Scott Spuler, Matt
Hayman, and Tammy Weckwerth at the National Center for Atmospheric Research.
Additionally, we thank Raisa Lehtinen, Reijo Roininen, and Christoph Münkel at Vaisala and Rob
Newsom at Pacific Northwest National Laboratory for the collection of the DIAL dataset at the
SGP site. This article supports activities associated with COST (European Cooperation in Science
and Technology) Action CA18235 "PROBE" (http://www.probe-cost.eu). We would like to
thank Jason English for constructive comments on an earlier version of this manuscript. This
paper does not imply endorsement for any particular instrument, nor reflect the views or
official position of NOAA or the U.S. government.



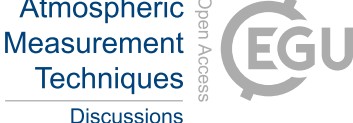

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






**Table 1:**
Important specifications of the instruments used in this paper

| Instrument | Specifications |
|---|---|
| MWR (HATPRO G4) | • 7 frequencies between 22.2 and 31.4 GHz<br>• 7 frequencies between 51.2 and 58.0 GHz<br>• Off-zenith data collected at elevations of 18° and 162°<br>• 1-s sky average, with elevation scans performed every 5 min; retrieval used single spectrum (both zenith and off-zenith) at desired time (e.g., close to sonde launch time<br>• Reference: Rose et al. 2005 |
| AERI | • 324 wavenumbers in these intervals: 612-618, 624-660, 674-713, 713-722, 538-588, 860.1-864.0, 872.2-877.5, 898.2-905.4 $cm^{-1}$<br>• 15-s sky average every 30-s; retrieval used single spectrum at desired time (e.g., close to sonde launch time)<br>• Principal component noise filter used to reduce random error (Turner et al. 2006)<br>• Reference: Knuteson et al. 2004 a,b |
| nDIAL | • Narrowband DIAL, transmitting at 830 nm<br>• Temporal resolution: 1-min<br>• Vertical resolution: 75-m<br>• Minimum height: 500 m; Maximum height was approx. 3 km (typical)<br>• Telescope receiver area (far field): 935 $cm^2$<br>• Average transmitted pulse power: 5 μJ pulses at 9 kHz (45 mW)<br>• Reference: Spuler et al. 2015; Weckwerth et al. 2016 |
| vDIAL | • Broadband DIAL, transmitting at 911 nm<br>• Temporal resolution: 20-min<br>• Vertical resolution: variable from 100 m at 100 m AGL to 200 m at 1 km<br>• Minimum height: 50 m; Maximum height was approx. 1 km (typical)<br>• Telescope receiver area (far field): 615 $cm^2$<br>• Average transmitted pulse power: 5.5 μJ pulses at 8 kHz (44 mW)<br>• Reference: Newsom et al. 2020 |







**Table 2**: Average uncertainty values (derived from $S_{op}$) at three levels for temperature and
humidity for the different instrument combinations used in this study.  The passive-only
retrievals are highlighted in gray, whereas the active+passive are in white.  The values in
parentheses at 3 km show the 10th and 90th percentile at that height, thereby providing a
measure of the amount of variability in these statistics for each retrieval.

| | Temperature Uncertainty [°C] | | | Water Vapor Uncertainty [g kg⁻¹] | | |
|---|---|---|---|---|---|---|
| | 500 m | 1000 m | 3000 m | 500 m | 1000 m | 3000 m |
| MWRz-only | 1.07 | 1.58 | 1.36 (1.34,1.36) | 1.11 | 1.35 | 0.87 (0.83,0.87) |
| MWRzo-only | 1.06 | 1.49 | 1.36 (1.34,1.36) | 1.11 | 1.34 | 0.87 (0.82,0.87) |
| AERI-only | 0.56 | 0.87 | 0.97 (0.86,1.22) | 0.73 | 1.01 | 0.96 (0.82,1.07) |
| AERI+MWRz | 0.56 | 0.86 | 0.94 (0.84,1.29) | 0.69 | 0.97 | 0.71 (0.64,0.78) |
| MWRz+nDIAL | 0.97 | 1.35 | 1.32 (1.28,1.35) | 0.73 | 0.67 | 0.68 (0.47,0.85) |
| MWRzo+nDIAL | 0.97 | 1.29 | 1.31 (1.27,1.35) | 0.73 | 0.66 | 0.68 (0.46,0.84) |
| AERI+nDIAL | 0.51 | 0.75 | 0.91 (0.81,1.22) | 0.57 | 0.62 | 0.74 (0.49,1.05) |
| AERI+MWRz+nDIAL | 0.51 | 0.75 | 0.91 (0.82,1.16) | 0.55 | 0.61 | 0.60 (0.42,0.75) |
| AERI-only (SGP) | 0.36 | 0.60 | 1.02 (0.82,1.41) | 0.65 | 1.00 | 1.17 (0.90,1.45) |
| AERI+vDIAL (SGP) | 0.35 | 0.57 | 1.01 (0.80,1.39) | 0.39 | 0.68 | 1.10 (0.81,1.42) |


**Table 3**: Average cDFS values at three levels for temperature and humidity for the different
instrument combinations used in this study.  The passive-only retrievals are highlighted in gray,
whereas the active+passive are in white. The values in parentheses at 3 km show the 10th and
90th percentile at that height, thereby providing a measure of the amount of variability in these
statistics for each retrieval.

| | Temperature cDFS value [unitless] | | | Water vapor cDFS value [unitless] | | |
|---|---|---|---|---|---|---|
| | 500 m | 1000 m | 3000 m | 500 m | 1000 m | 3000 m |
| MWRz-only | 1.51 | 1.82 | 2.15 (2.15,2.16) | 0.94 | 1.14 | 1.92 (1.71,2.03) |
| MWRzo-only | 1.85 | 2.22 | 2.57 (2.56,2.59) | 0.94 | 1.13 | 1.92 (1.71,2.03) |
| AERI-only | 3.87 | 4.55 | 5.50 (5.02,5.66) | 1.45 | 1.83 | 2.70 (1.88,3.41) |
| AERI+MWRz | 3.89 | 4.58 | 5.56 (5.15,5.66) | 1.53 | 1.97 | 3.17 (2.70,3.81) |
| MWRz+nDIAL | 1.51 | 1.82 | 2.16 (2.14,2.20) | 1.11 | 2.62 | 6.23 (1.97,9.44) |
| MWRzo+nDIAL | 1.84 | 2.20 | 2.57 (2.54,2.61) | 1.10 | 2.61 | 6.22 (1.99,9.41) |
| AERI+nDIAL | 3.87 | 4.52 | 5.48 (5.25,5.63) | 1.67 | 3.25 | 7.00 (2.80,10.14) |
| AERI+MWRz+nDIAL | 3.87 | 4.52 | 5.49 (5.25,5.63) | 1.71 | 3.28 | 7.21 (3.21,10.15) |
| AERI-only (SGP) | 4.80 | 5.53 | 6.58 (5.36,7.16) | 1.72 | 2.08 | 2.97 (1.90,3.83) |
| AERI+vDIAL (SGP) | 4.82 | 5.53 | 6.64 (5.45,7.13) | 2.54 | 4.17 | 5.50 (2.42,8.40) |


**Figures**:

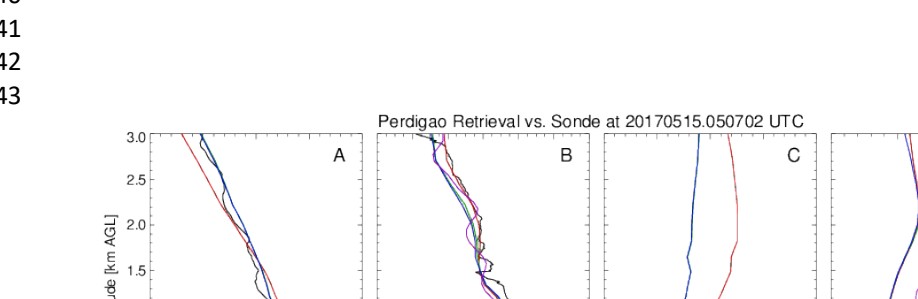

Fig 1: The retrieved profiles of temperature (A) and water vapor (B), with the uncertainties in these profiles (panels C and D, respectively), for the passive-only retrievals with the MWRzo only (red), AERI only (green), and AERI+MWRzo (blue) on 05:07 UTC on 15 May 2017 during Perdigao. The collocated radiosonde temperature and water vapor profiles are shown in black in (A) and (B), respectively. The water vapor observed by the DIAL and its uncertainty are included in the figure, although it is not used in any of these retrievals.

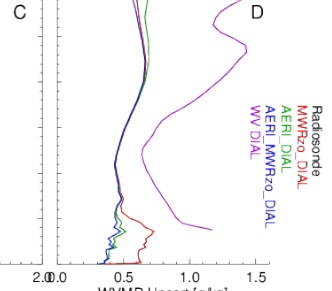

Fig 2: Same as Fig 1, except that the retrievals combine active and passive data with the MWRzo+DIAL (red), AERI+DIAL (green), and AERI+MWRzo+DIAL (blue). The water vapor observed by the DIAL and its uncertainty are included in the retrievals. See text for more details.




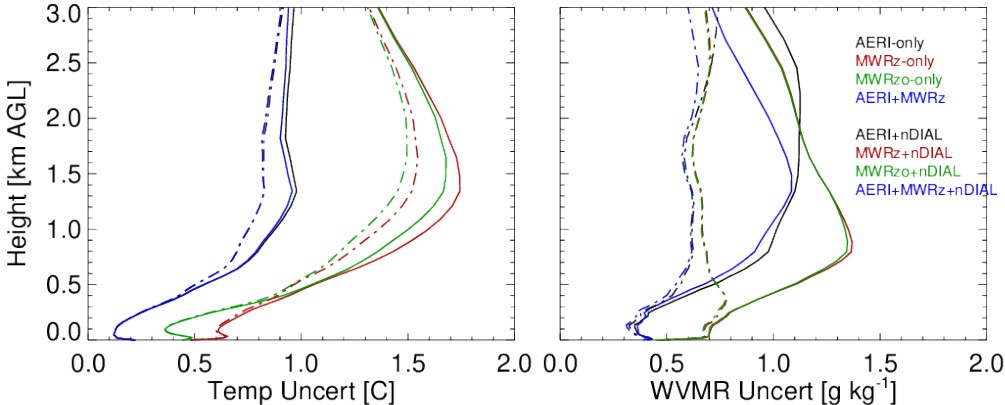

Fig 3: The mean uncertainty in temperature (left) and water vapor mixing ratio (right) for passive-only (solid lines) and active+passive (broken lines) retrievals during Perdigao.


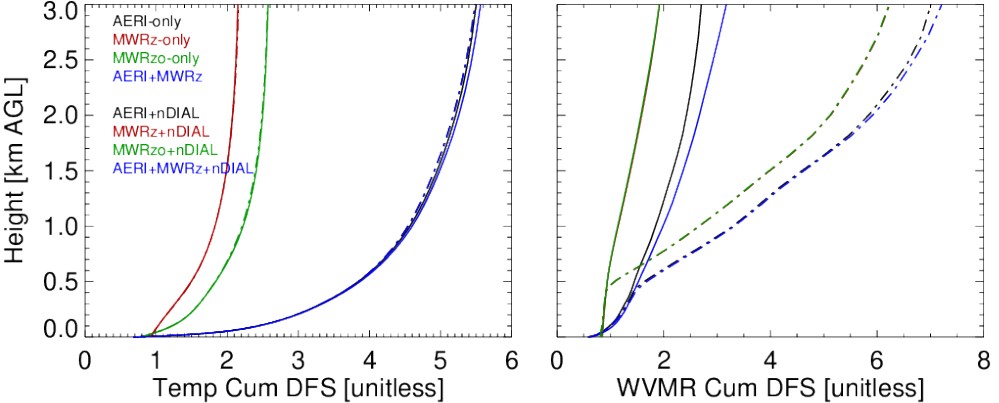

Fig 4: The mean cumulative degrees of DFS for temperature (left) and water vapor mixing ratio (right) for passive-only (solid lines) and active+passive (broken lines) retrievals during Perdigao. Note that the water vapor cumulative DFS profiles for MWRz and MWRzo retrievals are virtually identical (see Table 3) and hence overlap.





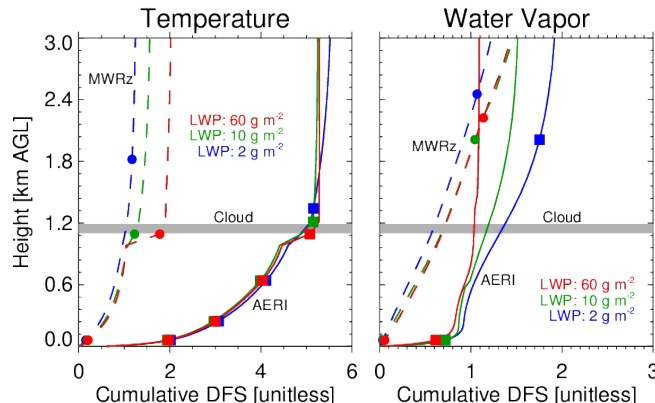

Fig 5: Profiles of cumulative degrees of freedom of signal from MWRz-only (dashed curves with dots) and AERI-only (solid curves with squares) temperature (left) and water vapor (right) retrievals for three samples between 03:00 and 05:00 UTC on 27 May 2017 during Perdigao. The different colors correspond to different LWP path values in the overhead cloud, whose height is indicated with the horizontal gray bar. The solid symbols indicate heights that would be assimilated, if the first level started at 50 m AGL and each level was separated by a unit of DFS. See the text for more details.


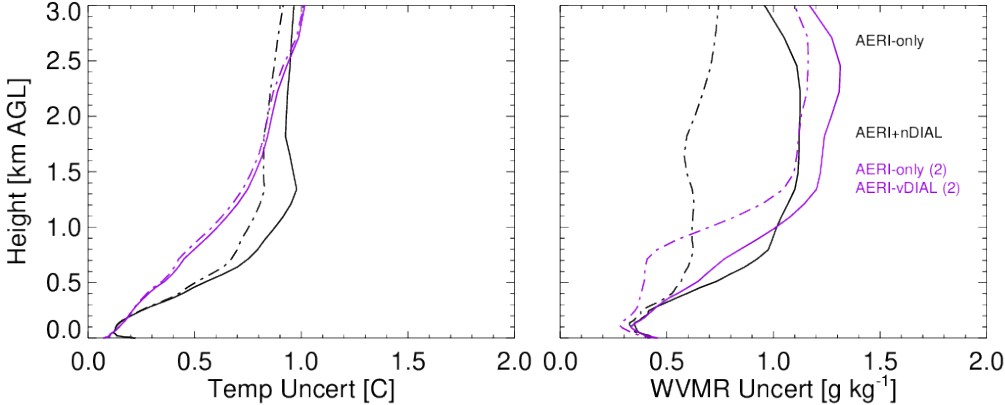

Fig 6: The mean uncertainty in temperature (left) and water vapor mixing ratio (right) for AERI-only (solid lines) and AERI+xDIAL (broken lines) retrievals during Perdigao (black) and SGP (purple), where the former used nDIAL data and the latter used vDIAL data. Note that different priors were used for the two locations; this impact is seen in the AERI-only retrievals as the noise levels of the two AERIs were similar.
