# Peer review of "Ground-based Temperature and Humidity Profiling: Combining Active and Passive Remote Sensors David D. Turner1 and Ulrich Löhnert2 1 NOAA / OAR / Global Systems Laboratory 2 University of Cologne / Institute of Geophysics and Meteorology Submitted 31 Au"

_Atmospheric Measurement Techniques, 2020_

## Referee Comment (RC1) · Anonymous Referee #1 · 19 Oct 2020

Turner and Ulrich Löhnert

**Anonymous Referee #1**

Summary:

This paper evaluates the synergies between ground-based infrared, microwave, and
WV-DIAL measurements to constrain boundary layer thermodynamic profiles. The fo-
cus of the paper is on the additional information contributed by the DIAL as these sys-
tems are rapidly advancing and will soon be commercially available. Optimal estimation
retrievals from SGP and the Perdigao field campaign are evaluated and compared. The
MW instrument is found to add little additional information above the infrared, with the
exception of a small contribution to the water vapor retrieval above $\sim$2 km. In contrast,
the DIAL adds significant information to the derived water vapor profile, but also helps

add information to the temperature retrieval, presumably by constraining the cross-talk between temperature and water vapor sensitivity in the passive observations.

The paper is clearly presented and the optimal estimation methodology is appropriate to address the issues of information content. I have only one major request of the authors below and a handful of minor comments.

Major comments:

The paper focuses entirely on the retrieval diagnostics (error variance, degrees of freedom, etc.). There is no direct validation of the retrieval itself. I would ask that the authors compare the retrieved profiles to the available radiosondes in a statistical manner. For example, does the observed difference between the retrieved profiles and the radiosondes have similar variance/covariance as the optimal estimation estimate. Are the retrievals biased in any systematic way? If there are biases or the estimated covariances are different than the observed validation, what implications would that have on your theoretical results and the measurement utility.

Minor Comments:

Line 161 and line 516: 'coadded' – is this a common terminology? I Infer that this is incoherent averaging but am unaware of this terminology.

Lines 280 – 289: I can't reconcile lines 280-282 which state the Perdigao had a DIAL and line 290 that state that the vDIAL was not part of the Perdigao campaign. Am I missing something or is this misstated?

Figure 1: I find it useful to add the a-priori mean profiles to these kinds of plots. For example, I would like to know if the a-priori includes the inversion or if the remote sensors are able to add that information.

Lines 599-607: It would be appropriate here to mention the PBL targeted observable from the decadal survey and the NASA incubation activities for a PBL mission, which will likely be composed of similar instruments.

---

## Referee Comment (RC2) · Anonymous Referee #2 · 9 Nov 2020

The manuscript addresses the critical problem of remotely sensing thermodynamic profiles within the planetary boundary layer and focuses on analyzing the synergies between passive and active ground-based instrument technologies. These include passive infrared and microwave profilers, as well as state-of-the-art differential absorption lidar (DIAL) systems that will soon be commercially available. The authors implement a combined retrieval algorithm that ingests previously retrieved water vapor profiles from the DIAL systems, and leverages these to improve optimal-estimation-based retrievals for the passive infrared and microwave systems. Two different DIAL instruments are analyzed, one from the company Vaisala and one from NCAR, using data from different measurement campaigns/locations (ARM SGP and the Perdigao

campaign in Portugal) with very different a priori profile distributions. The authors account for these important differences by assessing the impact of DIAL observations relative to the AERI instrument only retrievals, thus reducing the impact of the different climatologies in the different measurement locations. Impressive improvements in retrieval precision are reported for the passive systems, with reductions of up to 50%, and it is shown that the majority of the thermodynamic information in the PBL comes from the AERI and DIAL instruments. Furthermore, the authors demonstrate a meaningful reduction in temperature uncertainty that comes from including the DIAL observations of water vapor only.

In general the paper is very well written with sufficient mathematical detail and reasoning to support the main conclusions. I have a few important comments that I would ask the authors to address that I think will benefit the manuscript, as well as some minor comments.

Major comment #1:

I echo the other reviewer's comment that the manuscript lacks a discussion of accuracy in addition to the extensive precision discussions. How does combining the passive and active observations impact the accuracy of the retrievals (e.g. as compared with radiosonde profiles) in an ensemble/statistical sense (I recognize that there is one radiosonde profile plotted in figs 1 and 2)?

Major comment #2:

It is unclear how fundamentally "synergistic" these observations are. Specifically, I ask the authors to explore the difference between the combined retrieval implemented in this work, and the results that you would get if you simply fused the individual observations after performing passive-only and active-only retrievals (e.g. a weighted average of humidity profiles). In performing such a weighted average, of course one needs to be careful to incorporate the entire passive retrieval covariance matrix. I think showing a marked improvement from implementing a combined retrieval vs. simply fusing the

observations will be clear evidence of synergy. I will note that one example of synergy in this work is the reduction in temperature uncertainty from including active observations of water vapor only. However, as mentioned in Major Comment #1 the impact on accuracy is still an open question since there is not an analysis comparing retrievals vs. radiosondes.

Major comment #3:

The main thrusts of the paper seem to depend on the viewpoint from which you discuss the synergy. i.e. if you look at the problem as "how does adding DIAL observations improve the passive retrievals?" you may see a huge improvement in resolution and precision. However, if you look at it as "How much better do the DIAL retrievals become when adding passive observations?" the gains may be less substantial. I think this should be addressed in the paper.

Minor comments:

1. Line 273: These are not necessarily the only two options for using DIAL observations. Of course, modeling the backscattered energy is not realistically feasible, because you do not know the aerosol distribution. However, one could view the fundamental measurement of a DIAL as the differential optical depth between range bins. I wonder why you do not use this quantity as your DIAL element of the observation vector? If so, the simultaneous retrieval of temperature and inferred pressure allow for the absorption cross sections to be computed as part of the retrieval, and thus the retrieved DIAL water vapor concentration profile would be consistent with the temperature profile retrieved by AERI.

2. Line 341-344: I agree that the second question listed here is of considerable value. However, for the first question posed I raise my same point from above. This seems to assume that the necessary way to view the observational problem is from the perspective of DIAL improving AERI vs. MW retrievals. But what about the other way around? How do AERI and MW improve the DIAL retrievals if at all?

3. Figs 3: I think the DIAL-only average uncertainty profiles should be included on this plot.

4. Lines 438-442: This is an impressive improvement to the passive-only retrievals. It would be helpful to also state what the uncertainty reduction is compared to DIAL-only retrievals.

Very minor comments:

1. Tables 2 and 3: Are uncertainty values reported in the hundredths of a degree C and g/kg water vapor significant? My suggestion would be to use 2 significant digits.

1. Line 166: "were" should be "where".

3. Line 597: Should it be "virtual temperature profile observations" instead of "virtual temperature profiles observations"?

───────────────────────────

---

## Author Comment (AC1) · 16 Feb 2021

Reviewers comments are in black text; *our replies are in blue italics*.

**Anonymous Referee #1**

Summary:

This paper evaluates the synergies between ground-based infrared, microwave, and WV-DIAL measurements to constrain boundary layer thermodynamic profiles. The focus of the paper is on the additional information contributed by the DIAL as these systems are rapidly advancing and will soon be commercially available. Optimal estimation retrievals from SGP and the Perdigao field campaign are evaluated and compared. The MW instrument is found to add little additional information above the infrared, with the exception of a small contribution to the water vapor retrieval above ~2 km. In contrast, the DIAL adds significant information to the derived water vapor profile, but also helps add information to the temperature retrieval, presumably by constraining the cross-talk between temperature and water vapor sensitivity in the passive observations.

The paper is clearly presented and the optimal estimation methodology is appropriate to address the issues of information content. I have only one major request of the authors below and a handful of minor comments.

Major comments:

The paper focuses entirely on the retrieval diagnostics (error variance, degrees of freedom, etc.). There is no direct validation of the retrieval itself. I would ask that the authors compare the retrieved profiles to the available radiosondes in a statistical manner. For example, does the observed difference between the retrieved profiles and the radiosondes have similar variance/covariance as the optimal estimation estimate. Are the retrievals biased in any systematic way? If there are biases or the estimated co-variances are different than the observed validation, what implications would that have on your theoretical results and the measurement utility.

*To address this concern, we have added a new section with a new figure that describes the bias profiles for temperature and humidity from the various retrievals.*

Minor Comments:

Line 161 and line 516: 'coadded' – is this a common terminology? I infer that this is incoherent averaging but am unaware of this terminology.

*It is a common term in lidar remote sensing. It means to add photons from multiple laser shots as a function of range.*

Lines 280 – 289: I can't reconcile lines 280-282 which state the Perdigao had a DIAL and line 290 that state that the vDIAL was not part of the Perdigao campaign. Am I missing something or is this misstated?

*The NCAR water vapor DIAL (nDIAL) was deployed during Perdigao, and this was the first campaign that had an AERI, multi-channel MWR, and DIAL all collocated. However, there are currently no plans nDIAL commercially available, but the Vaisala DIAL (vDIAL) will soon be available commercially. Thus, we wanted to evaluate the impact of combining the nDIAL with the AERI and MWR data so we had to use Perdigao, and also to demonstrate the impact of the vDIAL (which has different performance characteristics) which required that we use the SGP dataset.*

Figure 1: I find it useful to add the a-priori mean profiles to these kinds of plots. For example, I would like to know if the a-priori includes the inversion or if the remote sensors are able to add that information.

*We added the prior profiles used in the retrieval as dotted black lines in panels A and B for figures 1 and 2.*

Lines 599-607: It would be appropriate here to mention the PBL targeted observable from the decadal survey and the NASA incubation activities for a PBL mission, which will likely be composed of similar instruments.

*Good suggestion: this was added along with a reference to the Decadal Survey*

---

## Author Comment (AC2) · 16 Feb 2021

Reviewers comments are in black text; *our replies are in blue italics*.

**Anonymous Referee #2**

The manuscript addresses the critical problem of remotely sensing thermodynamic profiles within the planetary boundary layer and focuses on analyzing the synergies between passive and active ground-based instrument technologies. These include passive infrared and microwave profilers, as well as state-of-the-art differential absorption lidar (DIAL) systems that will soon be commercially available. The authors implement a combined retrieval algorithm that ingests previously retrieved water vapor profiles from the DIAL systems, and leverages these to improve optimal-estimation-based retrievals for the passive infrared and microwave systems. Two different DIAL instruments are analyzed, one from the company Vaisala and one from NCAR, using data from different measurement campaigns/locations (ARM SGP and the Perdigao campaign in Portugal) with very different a priori profile distributions. The authors account for these important differences by assessing the impact of DIAL observations relative to the AERI instrument only retrievals, thus reducing the impact of the different climatologies in the different measurement locations. Impressive improvements in retrieval precision are reported for the passive systems, with reductions of up to 50%, and it is shown that the majority of the thermodynamic information in the PBL comes from the AERI and DIAL instruments. Furthermore, the authors demonstrate a meaningful reduction in temperature uncertainty that comes from including the DIAL observations of water vapor only.

In general the paper is very well written with sufficient mathematical detail and reasoning to support the main conclusions. I have a few important comments that I would ask the authors to address that I think will benefit the manuscript, as well as some minor comments.

Major comment #1:

I echo the other reviewer's comment that the manuscript lacks a discussion of accuracy in addition to the extensive precision discussions. How does combining the passive and active observations impact the accuracy of the retrievals (e.g. as compared with radiosonde profiles) in an ensemble/statistical sense (I recognize that there is one radiosonde profile plotted in figs 1 and 2)?

*This is a good point. We have added a new section and a new figure that shows the bias profiles compared to 169 radiosondes.*

Major comment #2:

It is unclear how fundamentally "synergistic" these observations are. Specifically, I ask the authors to explore the difference between the combined retrieval implemented in this work, and the results that you would get if you simply fused the individual observations after performing passive-only and active-only retrievals (e.g. a weighted average of humidity profiles). In performing such a weighted average, of course one needs to be careful to incorporate the entire passive retrieval covariance matrix. I think showing a marked improvement from implementing a combined retrieval vs. simply fusing the observations will be clear evidence of synergy. I will note that one

example of synergy in this work is the reduction in temperature uncertainty from including active observations of water vapor only. However, as mentioned in Major Comment #1 the impact on accuracy is still an open question since there is not an analysis comparing retrievals vs. radiosondes.

*The optimal estimation method allows the observations from the various instruments and the data from the prior to be blended in a well described manner to provide the solution that satisfies all of the observations within their uncertainties. We have referenced several papers that shows this, and in particular the seminal work by Clive Rodgers (2000). We believe that this method is indeed "synergistically" retrieving the best profile, assuming that the systematic errors in the observations is negligible.*

Major comment #3:

The main thrusts of the paper seem to depend on the viewpoint from which you discuss the synergy. i.e. if you look at the problem as "how does adding DIAL observations improve the passive retrievals?" you may see a huge improvement in resolution and precision. However, if you look at it as "How much better do the DIAL retrievals become when adding passive observations?" the gains may be less substantial. I think this should be addressed in the paper.

*The water vapor profiles from both of the DIALs were already characterized against radiosondes and other remote sensing observations in Weckwerth et al. (2016) and Newsom et al. (2020), both of which were discussed in this paper (around lines 180 for the nDIAL and lines 216 for the vDIAL). Thus, we did not feel it was necessary to reperform that type of characterization in this paper.*

Minor comments:

1. Line 273: These are not necessarily the only two options for using DIAL observations. Of course, modeling the backscattered energy is not realistically feasible, because you do not know the aerosol distribution. However, one could view the fundamental measurement of a DIAL as the differential optical depth between range bins. I wonder why you do not use this quantity as your DIAL element of the observation vector? If so, the simultaneous retrieval of temperature and inferred pressure allow for the absorption cross sections to be computed as part of the retrieval, and thus the retrieved DIAL water vapor concentration profile would be consistent with the temperature profile retrieved by AERI.

*This is true, and have modified the paper to indicate we could have used the differential optical depth as the observed variable from the DIALs. However, as the data product provided by both lidars is the water vapor concentration, we would have had to backwards derive the differential optical depth; so we just used the provided variable as the observation for this study.*

2. Line 341-344: I agree that the second question listed here is of considerable value. However, for the first question posed I raise my same point from above. This seems to assume that the necessary way to view the observational problem is from the perspective of DIAL improving AERI vs. MW retrievals. But what about the other way around? How do AERI and MW improve the DIAL retrievals if at all?

*Both DIALs only provide a partial profile of water vapor. The useful nDIAL range starts at 500 m above ground level, and the vDIAL observations seldom reach above 1 km. These shortcomings were*

*included in sections 2.3 and 2.4 that describes these lidars.  Thus, the use of the passive remote sensors with the DIAL data allows these shortcomings to be overcome, while still providing retrievals that are consistent with the DIAL profiles within the DIAL's errors.*

3. Figs 3: I think the DIAL-only average uncertainty profiles should be included on this plot.

*Excellent suggestion – the mean DIAL uncertainty was added to Fig 3.*

4. Lines 438-442: This is an impressive improvement to the passive-only retrievals. It would be helpful to also state what the uncertainty reduction is compared to DIAL-only retrievals.

*Good idea – we have added the lidar's water vapor uncertainty to Fig 3 and added a sentence to the end of section 4.2 to point this out.*

Very minor comments:

1.  Tables 2 and 3: Are uncertainty values reported in the hundredths of a degree C and g/kg water vapor significant? My suggestion would be to use 2 significant digits.

*Good idea: we updated the two tables accordingly*

2.  Line 166: "were" should be "where".

*Updated*

3. Line 597: Should it be "virtual temperature profile observations" instead of "virtual temperature profiles observations"?

*Updated*

---

## Author Response (AR2)

Replies to the comments from the Associate Editor:

Line 43: Replace 'IR instrument' with 'the IR instrument'.
Line 60: Replace 'in the thermodynamic profiling' with 'in thermodynamic profiling'.
Line 65: Replace 'atmosphere' with 'the atmosphere'.
Line 89: Replace 'are or will likely soon become' with 'are, or will likely soon, become'.
Lines 96-97: Replace 'height dependent' with 'height-dependent'.
Lines 122-123: Swap the order of the citations.
Lines 162, 542, 571: Although the terminology 'co-added' may be common in some parts of the lidar community, this manuscript is intended for a wider audience that is unlikely to have come across this term. I recommend either explaining this term in the first usage, or using a term such as 'photon accumulation'.
Line 178: Replace 'have' with 'has'.
Line 190: Replace 'this ceilometer is used' with 'a ceilometer used'.
Line 220: Remove 'the'.
Line 227: Replace 'a priori' with 'an a priori'.
Line 231: Replace 'satisfies' with 'satisfy'.
Line 249: Replace 'on' with 'of'.
Line 273: Replace 'the Eq 1,' with 'Eq 1,'.
Line 293: Replace 'are using' with 'use'.
Line 297: Replace 'the SGP' with 'the'.
Line 333: Replace 'like' with 'similar to'.
Line 358: Remove 'channels'.
Line 409: Replace 'the' with 'a'.
Line 456: Replace 'uses' with 'use'.
Line 459: Replace 'illustrate the' with 'illustrate that the'.
Lines 459-467 and Figs 3 and 4: You use MWRRe and MWRe-only in the text and in the figure legends. I presume these should be MWRzo and MWRzo-only?
Line 480: Replace 'used a single elevation angle to the' with either 'included one elevation angle for' or 'included one additional elevation angle for'.
Line 481: Replace 'increase' with 'increase of'.

We agreed with every of these suggestions, and modified the text as suggested. Note that we replaced figures 3 and figure 4 with images that had "MWRzo" in the keys (instead of the "MWRe" that was accidentally used before).